# Homogenous Granulation and Its Epsilon Variant [†]

**Krzysztof Ropiak** [‡] and **Piotr Artiemjew** [‡,*]

Faculty of Mathematics and Computer Science, University of Warmia and Mazury in Olsztyn, 10-710 Olsztyn, Poland; kropiak@matman.uwm.edu.pl

* Correspondence: artem@matman.uwm.edu.pl

† Extended version of paper "A Study in Granular Computing: homogenous granulation" presented at the 24rd International Conference on Information and Software Technologies (ICIST 2018), Vilnius, Lithuania, 5–6 October 2018.

‡ These authors contributed equally to this work.

**Abstract:** In the era of Big data, there is still place for techniques which reduce the data size with maintenance of its internal knowledge. This problem is the main subject of research of a family of granulation techniques proposed by Polkowski. In our recent works, we have developed new, really effective and simple techniques for decision approximation, homogenous granulation and epsilon homogenous granulation. The real problem in this family of methods was the choice of an effective parameter of approximation for any datasets. It was resolved by homogenous techniques. There is no need to estimate the optimal parameters of approximation for these methods, because those are set in a dynamic way according to the data internal indiscernibility level. In this work, we have presented an extension of the work presented at ICIST 2018 conference. We present results for homogenous and epsilon homogenous granulation with the comparison of its effectiveness.

**Keywords:** homogenous granulation; Rough Sets; decision systems; classification

## 1. Introduction

Granular rough computing is one of the techniques used for decision system approximation. This method relies on knowledge granules which are formed from objects with selected, similar features. The main goal is to reduce the amount of data being used for classification or regression, maintaining internal knowledge of the decision system. In the era of processing large datasets these techniques can play a significant role. Basic granulation methods were proposed by Polkowski [1,2]. In the works of Artiemjew [3,4], Polkowski [1,2,5–8], and Polkowski and Artiemjew [9–14] we have presented standard granulation, concept dependent and layered granulation in the context of data reduction, missing values absorbtion and usage in the classification process.

Our motivation to perform this research was an idea to determine effective indiscernibility ratio of decision system approximation without its estimation. The ratio of approximation has influence on the original data size reduction. In our previous methods. we had to estimate this parameter reviewing the set of radii from 0 to 1. In the methods proposed in this work, we do not have to perform this operation. The ratio, for particular central object, is chosen in an automatic way, by extending it until the set of objects is homogenous in the sense of belonging to the decision class. Instead of performing granulation several times, depending on the number of attributes of the object, this process is performed only once, solving the problem of optimal radii search. Our results are showing reduction of training dataset size by up to 50 percent maintaining the internal knowledge at a satisfying level which was measured by efficiency of the classification process. The method is simple, has a square time complexity, $U^2$ main operations time the scalar $|A|$. $U$ is the set of objects of decision system, $A$ the set of conditional attributes.

In this work, we have described the results of our previous research, presented in detail in [15,16]. The results are prepared for nominal (Homogenous granulation) data and numerical data (Epsilon homogenous granulation). It is worth to mention that our new methods were implemented in really effective new ensemble model; see [17].

The paper has the following content. In Section 2 there is a theoretical background. In Sections 3 and 4 we present a description of a our granulation techniques. In Section 5 we present a description of a classifier used in the experimental part. In Section 6 there are the results of our experiments and the conclusion is presented in Section 7.

There are three basic steps of the granulation process. The granules are computed for each training object, then, the training dataset is covered using the selected strategy and in the last step, majority voting is being used to get granular reflection of the training system.

In the next section, we describe the first step of the mentioned procedure.

## 2. Granular Rough Inclusions

Some more theory about rough inclusions can be found in Polkowski [1,6,7,18,19], a detailed discussion may be found in Polkowski [8].

For given objects $u$ and $v$ from training decision system $U, A, d$, where $U$ is the universe of objects, $A$ the set of conditional attributes, and $d$ is the decision attribute. The standard rough inclusion $\mu$ is defined as

$$\mu(v, u, r) \Leftrightarrow \frac{|IND(u, v)|}{|A|} \geq r \tag{1}$$

where

$$IND(u, v) = \{a \in A : a(u) = a(v)\}, \tag{2}$$

The parameter $r$ is the *granulation radius* from the set $\{0, \frac{1}{|A|}, \frac{2}{|A|}, ..., 1\}$.

### 2.1. ε–Modification of the Standard Rough Inclusion

Given a parameter $\varepsilon$ valued in the unit interval $[0, 1]$, we define the set

$$Ind_\varepsilon(u, v) = \{a \in A : dist(a(u), a(v)) \leq \varepsilon\}, \tag{3}$$

and, we set

$$\mu_\varepsilon(v, u, r) \Leftrightarrow \frac{|Ind_\varepsilon(u, v)|}{|A|} \geq r \tag{4}$$

### 2.2. Covering of Universe of Training Objects

During the process of covering the objects of the training system are covered based on chosen strategy. Simple random choice was used in this experiment, because it is the most effective method among studied ones; see [14]).

The last step of the granulation process is shown in the next section.

### 2.3. Granular Reflections

In this step the granular reflections of the original training system are formed based on the granules from the found coverage. Each granule $g \in COV(U, \mu, r)$ from the coverage is finally represented by a single object which attributes are chosen using the Majority Voting ($MV$) strategy.

$$\{MV(\{a(u) : u \in g\}) : a \in A \cup \{d\}\} \tag{5}$$

*The granular reflection* of the decision system $D = (U, A, d)$ is the decision system $COV(U, \mu, r)$, the set of objects formed from granules.

$$v \in g_r^{cd}(u) \text{ if and only if } \mu(v, u, r) \text{ and } (d(u) = d(v)) \tag{6}$$

for a given rough (weak) inclusion $\mu$.

Detailed information about our new method of granulation is presented in the next section.

## 3. Homogenous Granulation

In this section we have a formal definition of the homogenous granulation process. In plain words, considering the set of samples from a decision system, we can try to lower the size of the system by searching for groups of objects similar in a fixed ratio. Having those sets (the granules), we can cover the original system searching for granules, which represent all the knowledge from original decision systems. In this particular method, we form the group of objects, which belong to the same decision class and have the lowest possible indiscernibility ratio. It means that the similarity of samples is as low as possible until they are in the same class. The granule according to this assumption can be defined as $g_{r_u}^{homogenous}$; see the equation below.

The granules are formed as follows,

$$g_{r_u}^{homogenous} = \{v \in U : |g_{r_u}^{cd}| - |g_{r_u}| = 0, \text{ for minimal } r_u \text{ fulfills the equation}\} \tag{7}$$

where

$$g_{r_u}^{cd} = \{v \in U : \frac{IND(u, v)}{|A|} \le r_u \text{ AND } d(u) = d(v)\}$$

and

$$g_{r_u} = \{v \in U : \frac{IND(u, v)}{|A|} \le r_u\}$$

$$r_u = \{\frac{0}{|A|}, \frac{1}{|A|}, ..., \frac{|A|}{|A|}\}$$

### 3.1. Simple Example of Homogenous Granulation

In the Table 1, we have exemplary training decision system, which we based on while computing homogenous granules defined in previous section. The decision system $(U_{trn}, B, d)$ is the set of resolved problems, useful in modelling the automatic decision process. $U_{trn}$ is the set of objects from $u_1$ until $u_{24}$, $B$ is the set of conditional attributes (description of samples) and contains values from $b_1$ until $b_{13}$. $d$ is a decision attribute, which contains the expert decision used for creating the model of the classification. In our case two possible classes exist: $d \in D = \{1, 2\}$. Lets explain the process of granules formation. For given object $u_1$, which belongs to class 1 we are looking for the objects from the same class starting from the identical objects (similar in degree 1) until the objects are indiscernible in smallest possible degree (are the least similar to $u_1$) and all of them are in class 1. In case we greatly lower the indiscernibility ratio, the objects $r$-indiscernible will not point on the decision class in an unambiguous way. In our example ratio $0.385 = \frac{5}{13}$ for granule $g_{0.385}(u_1)$ means that the set contain objects, which are identical with the central one ($u_1$) at least on 5 positions. For instance, object $u_1$ and $u_6$ have the following common descriptors: $a_1 = 0$, $a_6 = 0$, $a_7 = 2$, $a_7 = 2$, $a_9 = 1$ and $a_{13} = 3$. In the covering part we are looking for the set of granules, which represent each object from $U_{trn}$ at least once.

Considering training decision system from Table 1.

**Table 1.** Example of decision system $(U_{trn}, B, d)$.

|  | $b_1$ | $b_2$ | $b_3$ | $b_4$ | $b_5$ | $b_6$ | $b_7$ | $b_8$ | $b_9$ | $b_{10}$ | $b_{11}$ | $b_{12}$ | $b_{13}$ | $d$ |
|---|---|---|---|---|---|---|---|---|---|---|---|---|---|---|
| $u_1$ | 74.0 | 0.0 | 2.0 | 120.0 | 269.0 | 0.0 | 2.0 | 121.0 | 1.0 | 0.2 | 1.0 | 1.0 | 3.0 | 1 |
| $u_2$ | 65.0 | 1.0 | 4.0 | 120.0 | 177.0 | 0.0 | 0.0 | 140.0 | 0.0 | 0.4 | 1.0 | 0.0 | 7.0 | 1 |
| $u_3$ | 59.0 | 1.0 | 4.0 | 135.0 | 234.0 | 0.0 | 0.0 | 161.0 | 0.0 | 0.5 | 2.0 | 0.0 | 7.0 | 1 |
| $u_4$ | 53.0 | 1.0 | 4.0 | 142.0 | 226.0 | 0.0 | 2.0 | 111.0 | 1.0 | 0.0 | 1.0 | 0.0 | 7.0 | 1 |
| $u_5$ | 43.0 | 1.0 | 4.0 | 115.0 | 303.0 | 0.0 | 0.0 | 181.0 | 0.0 | 1.2 | 2.0 | 0.0 | 3.0 | 1 |
| $u_6$ | 46.0 | 0.0 | 4.0 | 138.0 | 243.0 | 0.0 | 2.0 | 152.0 | 1.0 | 0.0 | 2.0 | 0.0 | 3.0 | 1 |
| $u_7$ | 60.0 | 1.0 | 4.0 | 140.0 | 293.0 | 0.0 | 2.0 | 170.0 | 0.0 | 1.2 | 2.0 | 2.0 | 7.0 | 2 |
| $u_8$ | 63.0 | 0.0 | 4.0 | 150.0 | 407.0 | 0.0 | 2.0 | 154.0 | 0.0 | 4.0 | 2.0 | 3.0 | 7.0 | 2 |
| $u_9$ | 40.0 | 1.0 | 1.0 | 140.0 | 199.0 | 0.0 | 0.0 | 178.0 | 1.0 | 1.4 | 1.0 | 0.0 | 7.0 | 1 |
| $u_{10}$ | 48.0 | 1.0 | 2.0 | 130.0 | 245.0 | 0.0 | 2.0 | 180.0 | 0.0 | 0.2 | 2.0 | 0.0 | 3.0 | 1 |
| $u_{11}$ | 54.0 | 0.0 | 2.0 | 132.0 | 288.0 | 1.0 | 2.0 | 159.0 | 1.0 | 0.0 | 1.0 | 1.0 | 3.0 | 1 |
| $u_{12}$ | 71.0 | 0.0 | 3.0 | 110.0 | 265.0 | 1.0 | 2.0 | 130.0 | 0.0 | 0.0 | 1.0 | 1.0 | 3.0 | 1 |
| $u_{13}$ | 70.0 | 1.0 | 4.0 | 130.0 | 322.0 | 0.0 | 2.0 | 109.0 | 0.0 | 2.4 | 2.0 | 3.0 | 3.0 | 2 |
| $u_{14}$ | 56.0 | 1.0 | 3.0 | 130.0 | 256.0 | 1.0 | 2.0 | 142.0 | 1.0 | 0.6 | 2.0 | 1.0 | 6.0 | 2 |
| $u_{15}$ | 59.0 | 1.0 | 4.0 | 110.0 | 239.0 | 0.0 | 2.0 | 142.0 | 1.0 | 1.2 | 2.0 | 1.0 | 7.0 | 2 |
| $u_{16}$ | 64.0 | 1.0 | 1.0 | 110.0 | 211.0 | 0.0 | 2.0 | 144.0 | 1.0 | 1.8 | 2.0 | 0.0 | 3.0 | 1 |
| $u_{17}$ | 67.0 | 1.0 | 4.0 | 120.0 | 229.0 | 0.0 | 2.0 | 129.0 | 1.0 | 2.6 | 2.0 | 2.0 | 7.0 | 2 |
| $u_{18}$ | 51.0 | 0.0 | 3.0 | 120.0 | 295.0 | 0.0 | 2.0 | 157.0 | 0.0 | 0.6 | 1.0 | 0.0 | 3.0 | 1 |
| $u_{19}$ | 64.0 | 1.0 | 4.0 | 128.0 | 263.0 | 0.0 | 0.0 | 105.0 | 1.0 | 0.2 | 2.0 | 1.0 | 7.0 | 1 |
| $u_{20}$ | 57.0 | 0.0 | 4.0 | 128.0 | 303.0 | 0.0 | 2.0 | 159.0 | 0.0 | 0.0 | 1.0 | 1.0 | 3.0 | 1 |
| $u_{21}$ | 71.0 | 0.0 | 4.0 | 112.0 | 149.0 | 0.0 | 0.0 | 125.0 | 0.0 | 1.6 | 2.0 | 0.0 | 3.0 | 1 |
| $u_{22}$ | 53.0 | 1.0 | 4.0 | 140.0 | 203.0 | 1.0 | 2.0 | 155.0 | 1.0 | 3.1 | 3.0 | 0.0 | 7.0 | 2 |
| $u_{23}$ | 47.0 | 1.0 | 4.0 | 112.0 | 204.0 | 0.0 | 0.0 | 143.0 | 0.0 | 0.1 | 1.0 | 0.0 | 3.0 | 1 |
| $u_{24}$ | 58.0 | 1.0 | 3.0 | 112.0 | 230.0 | 0.0 | 2.0 | 165.0 | 0.0 | 2.5 | 2.0 | 1.0 | 7.0 | 2 |

Homogenous granules are formed as follows:

$g_{0.385}(u_1) = (u_1, u_6, u_{10}, u_{11}, u_{12}, u_{18}, u_{20})$,
$g_{0.462}(u_2) = (u_2, u_3, u_4, u_5, u_9, u_{23})$,
$g_{0.539}(u_3) = (u_2, u_3, u_5)$,
$g_{0.615}(u_4) = (u_4)$,
$g_{0.539}(u_5) = (u_3, u_5, u_{21}, u_{23})$,
$g_{0.462}(u_6) = (u_4, u_6, u_{16}, u_{20}, u_{21})$,
$g_{0.539}(u_7) = (u_7, u_{15}, u_{17})$,
$g_{0.462}(u_8) = (u_7, u_8, u_{13})$,
$g_{0.462}(u_9) = (u_2, u_4, u_9)$,
$g_{0.615}(u_{10}) = (u_{10})$,
$g_{0.385}(u_{11}) = (u_1, u_6, u_{11}, u_{12}, u_{20})$,
$g_{0.385}(u_{12}) = (u_1, u_{11}, u_{12}, u_{18}, u_{20})$,
$g_{0.615}(u_{13}) = (u_{13})$,
$g_{0.385}(u_{14}) = (u_{14}, u_{15}, u_{24})$,
$g_{0.615}(u_{15}) = (u_{15})$,
$g_{0.539}(u_{16}) = (u_{16})$,
$g_{0.539}(u_{17}) = (u_7, u_{15}, u_{17})$,
$g_{0.389}(u_{18}) = (u_1, u_2, u_6, u_{10}, u_{12}, u_{18}, u_{20}, u_{21}, u_{23})$,
$g_{0.615}(u_{19}) = (u_{19})$,
$g_{0.462}(u_{20}) = (u_1, u_6, u_{11}, u_{12}, u_{18}, u_{20})$,
$g_{0.462}(u_{21}) = (u_3, u_5, u_6, u_{21}, u_{23})$,
$g_{0.615}(u_{22}) = (u_{22})$,
$g_{0.462}(u_{23}) = (u_2, u_3, u_5, u_{21}, u_{23})$,
$g_{0.462}(u_{24}) = (u_7, u_{15}, u_{24})$,

We cover the universe of objects by random choice:

$g_{0.462}(u_2) = (u_2, u_3, u_4, u_5, u_9, u_{23})$,
$g_{0.539}(u_3) = (u_2, u_3, u_5)$,
$g_{0.462}(u_6) = (u_4, u_6, u_{16}, u_{20}, u_{21})$,
$g_{0.462}(u_8) = (u_7, u_8, u_{13})$,
$g_{0.385}(u_{12}) = (u_1, u_{11}, u_{12}, u_{18}, u_{20})$,
$g_{0.385}(u_{14}) = (u_{14}, u_{15}, u_{24})$,
$g_{0.539}(u_{17}) = (u_7, u_{15}, u_{17})$,
$g_{0.385}(u_{18}) = (u_1, u_2, u_6, u_{10}, u_{12}, u_{18}, u_{20}, u_{21}, u_{23})$,
$g_{0.615}(u_{19}) = (u_{19})$,
$g_{0.462}(u_{21}) = (u_3, u_5, u_6, u_{21}, u_{23})$,
$g_{0.615}(u_{22}) = (u_{22})$,

Final granular system is in Table 2.

**Table 2.** Granular decision system formed from Covering granules.

| | $b_1$ | $b_2$ | $b_3$ | $b_4$ | $b_5$ | $b_6$ | $b_7$ | $b_8$ | $b_9$ | $b_{10}$ | $b_{11}$ | $b_{12}$ | $b_{13}$ | $d$ |
|---|---|---|---|---|---|---|---|---|---|---|---|---|---|---|
| $g_{0.462}(u_2)$ | 65.0 | 1.0 | 4.0 | 120.0 | 177.0 | 0.0 | 0.0 | 140.0 | 0.0 | 0.4 | 1.0 | 0.0 | 7.0 | 1 |
| $g_{0.539}(u_3)$ | 65.0 | 1.0 | 4.0 | 120.0 | 177.0 | 0.0 | 0.0 | 140.0 | 0.0 | 0.4 | 2.0 | 0.0 | 7.0 | 1 |
| $g_{0.462}(u_6)$ | 53.0 | 0.0 | 4.0 | 142.0 | 226.0 | 0.0 | 2.0 | 111.0 | 1.0 | 0.0 | 2.0 | 0.0 | 3.0 | 1 |
| $g_{0.462}(u_8)$ | 60.0 | 1.0 | 4.0 | 140.0 | 293.0 | 0.0 | 2.0 | 170.0 | 0.0 | 1.2 | 2.0 | 3.0 | 7.0 | 2 |
| $g_{0.385}(u_{12})$ | 74.0 | 0.0 | 2.0 | 120.0 | 269.0 | 0.0 | 2.0 | 159.0 | 0.0 | 0.0 | 1.0 | 1.0 | 3.0 | 1 |
| $g_{0.385}(u_{14})$ | 56.0 | 1.0 | 3.0 | 130.0 | 256.0 | 0.0 | 2.0 | 142.0 | 1.0 | 0.6 | 2.0 | 1.0 | 7.0 | 2 |
| $g_{0.539}(u_{17})$ | 60.0 | 1.0 | 4.0 | 140.0 | 293.0 | 0.0 | 2.0 | 170.0 | 1.0 | 1.2 | 2.0 | 2.0 | 7.0 | 2 |
| $g_{0.385}(u_{18})$ | 71.0 | 0.0 | 4.0 | 120.0 | 269.0 | 0.0 | 2.0 | 121.0 | 0.0 | 0.0 | 1.0 | 0.0 | 3.0 | 1 |
| $g_{0.615}(u_{19})$ | 64.0 | 1.0 | 4.0 | 128.0 | 263.0 | 0.0 | 0.0 | 105.0 | 1.0 | 0.2 | 2.0 | 1.0 | 7.0 | 1 |
| $g_{0.462}(u_{21})$ | 59.0 | 1.0 | 4.0 | 112.0 | 234.0 | 0.0 | 0.0 | 161.0 | 0.0 | 0.5 | 2.0 | 0.0 | 3.0 | 1 |
| $g_{0.615}(u_{22})$ | 53.0 | 1.0 | 4.0 | 140.0 | 203.0 | 1.0 | 2.0 | 155.0 | 1.0 | 3.1 | 3.0 | 0.0 | 7.0 | 2 |

Exemplary visualization of granulation process is presented in Figure 1.

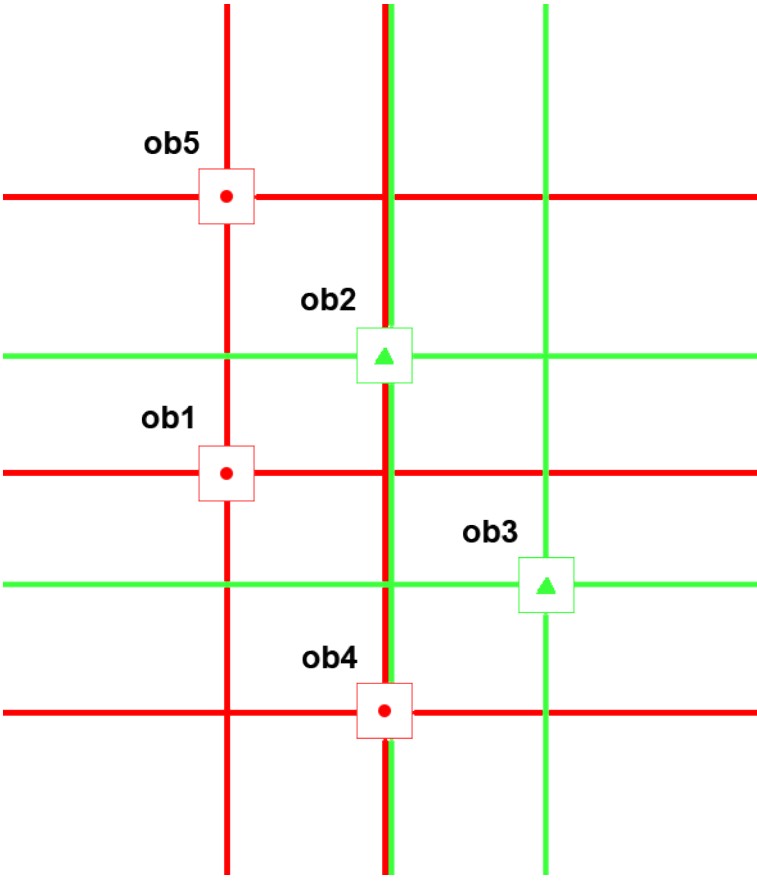

**Figure 1.** Simple demonstration of granulation for objects represented by the pairs of attributes. In the picture we have objects of two classes, circles and triangles. Granulating the decision system in homogenous way we can obtain $g_{0.5}(ob1) = \{ob1, ob5\}$, $g_1(ob2) = \{ob2\}$, $g_{0.5}(ob3) = \{ob3\}$, $g_1(ob4) = \{ob4\}$, $g_{0.5}(ob1) = \{ob5, ob1\}$. The set of possible radii is $\{\frac{0}{2}, \frac{1}{2}, \frac{2}{2}\}$.

## 4. Epsilon Variant of Homogenous Granulation

The only difference according to homogenous granulation described in Section 3.1 is the addition of the parameter $\varepsilon$, which allows us to use a floating point value in the process of granulation. The rest of the techniques are similar.

The method is defined in the following way,

$$g_{r_u}^{\varepsilon, homogenous} = \{v \in U : |g_{r_u}^{\varepsilon-cd}| - |g_{r_u}^{\varepsilon}| = 0, \text{ for minimal } r_u \text{ fulfills the equation}\} \qquad (8)$$

where

$$g_{r_u}^{\varepsilon, cd}(u) = \{v \in U : \frac{IND_\varepsilon(u, v)}{|A|} \le r_u \text{ AND } d(u) = d(v)\}$$

and

$$g_{r_u}^{\varepsilon}(u) = \{v \in U : \frac{IND_\varepsilon(u, v)}{|A|} \le r_u\}$$

$$r_u = \{\frac{0}{|A|}, \frac{1}{|A|}, ..., \frac{|A|}{|A|}\}$$

$$IND_\varepsilon(u, v) = \{a \in A : \frac{|a(u) - a(v)|}{max_a - min_a} \le \varepsilon\}$$

where $max_a$, $min_a$ are the maximal and minimal attribute values for $a \in A$ in the original dataset.

The metrics for epsilon granulation and classification are defined in Equations (9) and (10) respectively. The Hamming metric for symbolic data is placed in Equation (9). $\varepsilon$-normalized Hamming metric as modification for numerical values, for given $\varepsilon$ is in Equation (10).

$$d_H(u, v) = |\{a \in A : a(u) \neq a(v)\}|. \tag{9}$$

$$d_{H,\varepsilon}(u, v) = |\{a \in A : \frac{|a(u) - a(v)|}{max_a - min_a} > \varepsilon\}|. \tag{10}$$

Considering training decision system from Table 3 the hand example of $\varepsilon$ homogenous granulation is as follows.

**Table 3.** Training data system $(U_{trn}, A, d)$, (a sample from australian credit dataset), for $\varepsilon = 0.05$.

|          | $b_1$ | $b_2$ | $b_3$ | $b_4$ | $b_5$ | $b_6$ | $b_7$ | $b_8$ | $b_9$ | $b_{10}$ | $b_{11}$ | $b_{12}$ | $b_{13}$ | $d$ |   |
|----------|-------|-------|-------|-------|-------|-------|-------|-------|-------|----------|----------|----------|----------|-------|---|
| $u_1$    | 1     | 20.17 | 8.17  | 2     | 6     | 4     | 1.96  | 1     | 1     | 14       | 0        | 2        | 60       | 159   | 1 |
| $u_2$    | 1     | 34.92 | 5     | 2     | 14    | 8     | 7.5   | 1     | 1     | 6        | 1        | 2        | 0        | 1001  | 1 |
| $u_3$    | 1     | 58.58 | 2.71  | 2     | 8     | 4     | 2.415 | 0     | 0     | 0        | 1        | 2        | 320      | 1     | 0 |
| $u_4$    | 1     | 29.58 | 4.5   | 2     | 9     | 4     | 7.5   | 1     | 1     | 2        | 1        | 2        | 330      | 1     | 1 |
| $u_5$    | 0     | 19.17 | 0.58  | 1     | 6     | 4     | 0.585 | 1     | 0     | 0        | 1        | 2        | 160      | 1     | 0 |
| $u_6$    | 1     | 23.08 | 2.5   | 2     | 8     | 4     | 1.085 | 1     | 1     | 11       | 1        | 2        | 60       | 2185  | 1 |
| $u_7$    | 0     | 21.67 | 11.5  | 1     | 5     | 3     | 0     | 1     | 1     | 11       | 1        | 2        | 0        | 1     | 1 |
| $u_8$    | 1     | 27.83 | 1     | 1     | 2     | 8     | 3     | 0     | 0     | 0        | 0        | 2        | 176      | 538   | 0 |
| $u_9$    | 1     | 41.17 | 1.33  | 2     | 2     | 4     | 0.165 | 0     | 0     | 0        | 0        | 2        | 168      | 1     | 0 |
| $u_{10}$ | 1     | 41.58 | 1.75  | 2     | 4     | 4     | 0.21  | 1     | 0     | 0        | 0        | 2        | 160      | 1     | 0 |
| $u_{11}$ | 1     | 22.5  | 0.12  | 1     | 4     | 4     | 0.125 | 0     | 0     | 0        | 0        | 2        | 200      | 71    | 0 |
| $u_{12}$ | 1     | 33.17 | 3.04  | 1     | 8     | 8     | 2.04  | 1     | 1     | 1        | 1        | 2        | 180      | 18028 | 1 |
| $u_{13}$ | 1.234 | 22.08 | 11.46 | 2     | 4     | 4     | 1.585 | 0     | 0     | 0        | 1        | 2        | 100      | 1213  | 0 |
| $u_{14}$ | 0     | 58.67 | 4.46  | 2     | 11    | 8     | 3.04  | 1     | 1     | 6        | 0        | 2        | 43       | 561   | 1 |
| $u_{15}$ | 1     | 33.5  | 1.75  | 2     | 14    | 8     | 4.5   | 1     | 1     | 4        | 1        | 2        | 253      | 858   | 1 |
| $u_{16}$ | 0     | 18.92 | 9     | 2     | 6     | 4     | 0.75  | 1     | 1     | 2        | 0        | 2        | 88       | 592   | 1 |
| $u_{17}$ | 1     | 20    | 1.25  | 1     | 4     | 4     | 0.125 | 0     | 0     | 0        | 0        | 2        | 140      | 5     | 0 |
| $u_{18}$ | 1     | 19.5  | 9.58  | 2     | 6     | 4     | 0.79  | 0     | 0     | 0        | 0        | 2        | 80       | 351   | 0 |
| $u_{19}$ | 0     | 22.67 | 3.8   | 2     | 8     | 4     | 0.165 | 0     | 0     | 0        | 0        | 2        | 160      | 1     | 0 |
| $u_{20}$ | 1     | 17.42 | 6.5   | 2     | 3     | 4     | 0.125 | 0     | 0     | 0        | 0        | 2        | 60       | 101   | 0 |
| $u_{21}$ | 1     | 41.42 | 5     | 2     | 11    | 8     | 5     | 1     | 1     | 6        | 1        | 2        | 470      | 1     | 1 |
| $u_{22}$ | 1     | 20.67 | 1.25  | 1     | 8     | 8     | 1.375 | 1     | 1     | 3        | 1        | 2        | 140      | 211   | 0 |
| $u_{23}$ | 1     | 48.08 | 6.04  | 2     | 4     | 4     | 0.04  | 0     | 0     | 0        | 0        | 2        | 0        | 2691  | 1 |
| $u_{24}$ | 0     | 28.17 | 0.58  | 2     | 6     | 4     | 0.04  | 0     | 0     | 0        | 0        | 2        | 260      | 1005  | 0 |

The granules are computed below:

$g_0.571429(u_1) = (u_1),$

$g_0.5(u_2) = (u_2, u_4, u_{15}, u_{21}),$

$g_0.571429(u_3) = (u_3, u_9, u_{19}, u_{20}),$

$g_0.5(u_4) = (u_1, u_2, u_4, u_6, u_{21}),$

$g_0.5(u_5) = (u_5, u_{10}, u_{19}, u_{24}),$

$g_0.5(u_6) = (u_1, u_4, u_6),$

$g_0.5(u_7) = (u_7),$

$g_0.5(u_8) = (u_8, u_9, u_{11}, u_{17}),$

$g_0.642857(u_9) = (u_9, u_{10}, u_{11}, u_{17}, u_{19}, u_{20}),$

$g_0.642857(u_{10}) = (u_9, u_{10}, u_{19}),$

$g_0.642857(u_{11}) = (u_9, u_{11}, u_{17}, u_{19}, u_{20}),$

$g_0.642857(u_{12}) = (u_{12}),$

$g_0.571429(u_{13}) = (u_{13}),$

$g_0.428571(u_{14}) = (u_2, u_{14}, u_{16}, u_{21}),$

$g_0.5(u_{15}) = (u_2, u_{12}, u_{15}, u_{21})$,
$g_0.5(u_{16}) = (u_1, u_{14}, u_{16})$,
$g_0.642857(u_{17}) = (u_9, u_{11}, u_{17}, u_{20})$,
$g_0.642857(u_{18}) = (u_{18})$,
$g_0.571429(u_{19}) = (u_3, u_9, u_{10}, u_{11}, u_{17}, u_{19}, u_{20}, u_{24})$,
$g_0.642857(u_{20}) = (u_9, u_{11}, u_{17}, u_{19}, u_{20})$,
$g_0.5(u_{21}) = (u_2, u_4, u_{14}, u_{15}, u_{21})$,
$g_0.642857(u_{22}) = (u_{22})$,
$g_0.642857(u_{23}) = (u_{23})$,
$g_0.642857(u_{24}) = (u_{24})$,

Granules covering training system by random choice:

Covering granules: $g_0.5(u_2) = (u_2, u_4, u_{15}, u_{21})$,
$g_0.571429(u_3) = (u_3, u_9, u_{19}, u_{20})$,
$g_0.5(u_5) = (u_5, u_{10}, u_{19}, u_{24})$,
$g_0.5(u_6) = (u_1, u_4, u_6)$,
$g_0.5(u_7) = (u_7)$,
$g_0.5(u_8) = (u_8, u_9, u_{11}, u_{17})$,
$g_0.642857(u_{12}) = (u_{12})$,
$g_0.571429(u_{13}) = (u_{13})$,
$g_0.5(u_{16}) = (u_1, u_{14}, u_{16})$,
$g_0.642857(u_{18}) = (u_{18})$,
$g_0.642857(u_{20}) = (u_9, u_{11}, u_{17}, u_{19}, u_{20})$,
$g_0.5(u_{21}) = (u_2, u_4, u_{14}, u_{15}, u_{21})$,
$g_0.642857(u_{22}) = (u_{22})$,
$g_0.642857(u_{23}) = (u_{23})$,

Final approximation of training decision system is in Table 4:

**Table 4.** Granular decision system formed from Covering granules.

|  | $b_1$ | $b_2$ | $b_3$ | $b_4$ | $b_5$ | $b_6$ | $b_7$ | $b_8$ | $b_9$ | $b_{10}$ | $b_{11}$ | $b_{12}$ | $b_{13}$ | $d$ |  |
|---|---|---|---|---|---|---|---|---|---|---|---|---|---|---|---|
| $g_0.5(u_2)$ | 1 | 34.92 | 5 | 2 | 14 | 8 | 7.5 | 1 | 1 | 6 | 1 | 2 | 0 | 1001 | 1 |
| $g_0.571429(u_3)$ | 1 | 58.58 | 2.71 | 2 | 8 | 4 | 0.165 | 0 | 0 | 0 | 0 | 2 | 320 | 1 | 0 |
| $g_0.5(u_5)$ | 0 | 19.17 | 0.58 | 2 | 6 | 4 | 0.21 | 1 | 0 | 0 | 0 | 2 | 160 | 1 | 0 |
| $g_0.5(u_6)$ | 1 | 20.17 | 8.17 | 2 | 6 | 4 | 1.96 | 1 | 1 | 14 | 1 | 2 | 60 | 159 | 1 |
| $g_0.5(u_7)$ | 0 | 21.67 | 11.5 | 1 | 5 | 3 | 0 | 1 | 1 | 11 | 1 | 2 | 0 | 1 | 1 |
| $g_0.5(u_8)$ | 1 | 27.83 | 1.33 | 1 | 2 | 4 | 0.165 | 0 | 0 | 0 | 0 | 2 | 176 | 1 | 0 |
| $g_0.642857(u_{12})$ | 1 | 33.17 | 3.04 | 1 | 8 | 8 | 2.04 | 1 | 1 | 1 | 1 | 2 | 180 | 18028 | 1 |
| $g_0.571429(u_{13})$ | 1.234 | 22.08 | 11.46 | 2 | 4 | 4 | 1.585 | 0 | 0 | 0 | 1 | 2 | 100 | 1213 | 0 |
| $g_0.5(u_{16})$ | 0 | 20.17 | 8.17 | 2 | 6 | 4 | 1.96 | 1 | 1 | 14 | 0 | 2 | 60 | 561 | 1 |
| $g_0.642857(u_{18})$ | 1 | 19.5 | 9.58 | 2 | 6 | 4 | 0.79 | 0 | 0 | 0 | 0 | 2 | 80 | 351 | 0 |
| $g_0.642857(u_{20})$ | 1 | 22.5 | 1.33 | 2 | 4 | 4 | 0.165 | 0 | 0 | 0 | 0 | 2 | 168 | 1 | 0 |
| $g_0.5(u_{21})$ | 1 | 34.92 | 5 | 2 | 14 | 8 | 7.5 | 1 | 1 | 6 | 1 | 2 | 0 | 1001 | 1 |
| $g_0.642857(u_{22})$ | 1 | 20.67 | 1.25 | 1 | 8 | 8 | 1.375 | 1 | 1 | 3 | 1 | 2 | 140 | 211 | 0 |
| $g_0.642857(u_{23})$ | 1 | 48.08 | 6.04 | 2 | 4 | 4 | 0.04 | 0 | 0 | 0 | 0 | 2 | 0 | 2691 | 1 |

In the Figure 2 there is a simple visualization of granulation process.

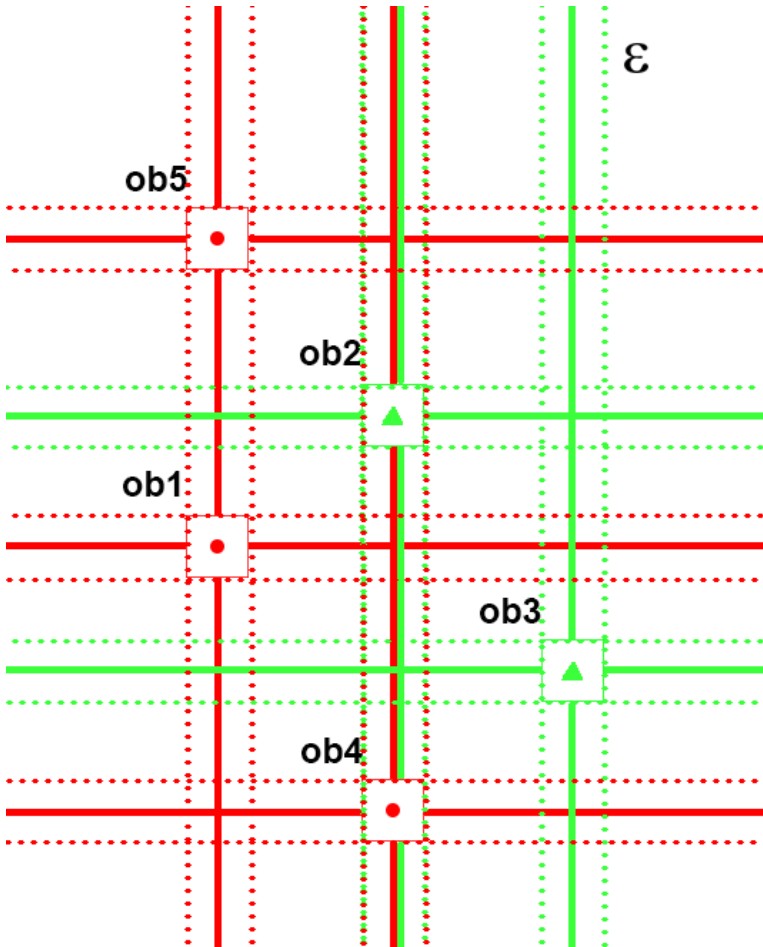

**Figure 2.** Exemplary toy demonstration for objects represented as pairs of attributes. We have two decision concepts: circles and rectangles. Epsilon homogenous granules can be $g_{0.5}^{\varepsilon}(ob1) = \{ob1, ob5\}$, $g_{1}^{\varepsilon}(ob2) = \{ob2\}$, $g_{0.5}^{\varepsilon}(ob3) = \{ob3\}$, $g_{1}^{\varepsilon}(ob4) = \{ob4\}$, $g_{0.5}^{\varepsilon}(ob1) = \{ob5, ob1\}$. The set of possible radii is $\{\frac{0}{2}, \frac{1}{2}, \frac{2}{2}\}$. The descriptors can be shifted in the range determined by $\varepsilon$ and still were treated as indiscernible.

## 5. Description of Classifier Used for Evaluation of the Granulation

A *kNN* classifier has been used in the experiments to verify the effectiveness of approximation. The procedure is as follows.

Step 1. The training granular decision system $(G_{r_{gran}}^{trn}, A, d)$ and the test decision system $(U_{tst}, A, d)$ are given, where $A$ is a set of conditional attributes, $d$ is the decision attribute, and $r_{gran}$ a granulation radius.

Step 2. Classification of test objects, by means of granules of training objects, is performed as follows.

For all conditional attributes $a \in A$, training objects $v \in G^{trn}$, and test objects $u \in U_{tst}$, we compute weights $w(u, v)$ based on the Hamming metric.

In the voting procedure of the *kNN* classifier, we use optimal $k$ estimated by CV5, details of the procedure are highlighted in the next section.

If the cardinality of the smallest training decision class is less than $k$, we apply the value for $k = |the\ smallest\ training\ decision\ class|$.

The test object $u$ is classified by means of weights computed for all training objects $v$. Weights are sorted in ascending order as,

$$w_1^{c_1}(u, v_1^{c_1}) \leq w_2^{c_1}(u, v_2^{c_1}) \leq \ldots \leq w_{|C_1|}^{c_1}(u, v_{|C_1|}^{c_1});$$

$$w_1^{c_2}(u, v_1^{c_2}) \leq w_2^{c_2}(u, v_2^{c_2}) \leq \ldots \leq w_{|C_2|}^{c_2}(u, v_{|C_2|}^{c_2});$$

$$\ldots$$

$$w_1^{c_m}(u, v_1^{c_m}) \leq w_2^{c_m}(u, v_2^{c_m}) \leq \ldots \leq w_{|C_m|}^{c_m}(u, v_{|C_m|}^{c_m}),$$

where $C_1, C_2, ..., C_m$ are all decision classes in the training set.

Based on computed and sorted weights, training decision classes vote by means of the following parameter, where $c$ runs over decision classes in the training set,

$$Concept\_weight_c(u) = \sum_{i=1}^{k} w_i^c(u, v_i^c). \tag{11}$$

Finally, the test object $u$ is classified into the class $c$ with a minimal value of $Concept\_weight_c(u)$.

After all test objects $u$ are classified, the quality parameter of *accuracy (acc)* is computed, according to the formula

$$acc = \frac{number\ of\ correctly\ classified\ objects}{number\ of\ classified\ objects}. \tag{12}$$

*Parameter Estimation in kNN Classifier*

In our experiments, we use the classical version of *kNN* classifier based on the Hamming metric. In the first step, we estimate the optimal $k$ based on $5\times$ CV5 cross-validation on the part of dataset. In the next step, we use the estimated value of $k$ in order to find $k$ nearest objects for each decision class and then voting is performed to select the decision. If the value of $k$ is larger than the smallest training decision class cardinality then $k$ value is equal to cardinality of this class.

In Table 5 we can see the estimated values of $k$ for all tested datasets. These values were chosen as optimal based on the experiments with various values of $k$ and results estimated by multiple $CV5$ operations.

**Table 5.** Estimated parameters for *kNN* based on $5 \times CV5$ cross–validation, data from UCI Repository [20].

| *Name* | *Optimal k* |
|---|---|
| *Australian-credit* | 5 |
| *Car Evaluation* | 8 |
| *Diabetes* | 3 |
| *German-credit* | 18 |
| *Heartdisease* | 19 |
| *Hepatitis* | 3 |
| *Nursery* | 4 |
| *SPECTF Heart* | 14 |

## 6. The Results of Experiments

To show the effectiveness of the new method, we have carried out a series of experiments with real data from University of Irvine Repository (see [20]). The reference classifier is *kNN* with Cross Validation 5 model. Data for experiments are listed in Table 6. The k parameter was evaluated in our previous works [14]. The list of optimal parameters of k is shown in Table 5. The single test consists of splitting the data into training and test set, where the training samples are granulated using our homogenous method. The results of the experiments are presented in Table 7. We have shown the comparable effectiveness of this new method in comparison with our best concept dependent granulation method; see Table 8. The new technique is significantly different from existing methods. Dynamic tuning of radius during granulation results with granules directed on decisions of their central objects. The radius is selected in automatic way during granulation process so there is no need to estimate optimal radius of granulation. The approximation level depends on objects

indiscernibility ratio in the particular decision classes. Epsilon variant; see Table 9 is fully comparable to the homogenous method and works more precisely for numerical data.

**Table 6.** Basic information about datasets-[20].

| Name | Attr Type | Attr no. | Obj no. | Class no. |
|---|---|---|---|---|
| Australian-credit | categorical, integer, real | 15 | 690 | 2 |
| Car Evaluation | categorical | 7 | 1728 | 4 |
| Diabetes | categorical, integer | 9 | 768 | 2 |
| German-credit | categorical, integer | 21 | 1000 | 2 |
| Heartdisease | categorical, real | 14 | 270 | 2 |
| Hepatitis | categorical, integer, real | 20 | 155 | 2 |
| Nursery | categorical | 9 | 12,960 | 5 |
| SPECTF Heart | integer | 45 | 267 | 2 |

**Table 7.** The result for dynamic granulation; $5 \times CV5$ method with *kNN* classifier; $acc\_5CV5 = $ *average accuracy*, $GS\_size = $ *granular decision system size*, $TRN\_size = $ *training set size*, $TRN\_reduction = $ *reduction in object number in training size*, $radii\_range = $ *spectrum of radii*.

| Name | acc | GS_size | TRN_size | TRN_reduction | radii_range |
|---|---|---|---|---|---|
| Australian-credit | 0.835 | 286.52 | 552 | 48.1% | $r_u \geq 0.5$ |
| Car Evaluation | 0.797 | 728.5 | 1382 | 47.3% | $r_u \geq 0.667$ |
| Diabetes | 0.653 | 488.9 | 614 | 20.4% | $r_u \geq 0.25$ |
| German-credit | 0.725 | 513.3 | 800 | 35.8% | $r_u \geq 0.6$ |
| Heartdisease | 0.833 | 120.5 | 216 | 44.2% | $r_u \geq 0.461$ |
| Hepatitis | 0.88 | 46.16 | 124 | 62.8% | $r_u \geq 0.579$ |
| Nursery | 0.607 | 9009.1 | 10368 | 13.1% | $r_u \geq 0.875$ |
| SPECTF Heart | 0.763 | 138.75 | 214 | 35.2% | $r_u \geq 0.068$ |

**Table 8.** Summary of results for *kNN* Classifier, granular and non granular case, *acc* = accuracy of classification, *red* = percentage reduction in object number, *r* = granulation radius, *method* = variant of Naive Bayes classifier, *nil.acc* = non granular case.

| Name | acc, red, r | nil.acc |
|---|---|---|
| Australian-credit | 0.851, 71.86, 0.571 | 0.855 |
| Car Evaluation | 0.865, 73.23, 0.833 | 0.944 |
| Diabetes | 0.616, 74.74, 0.25 | 0.631 |
| German-credit | 0.724, 59.85, 0.65 | 0.73 |
| Heartdisease | 0.83, 67.69, 0.538 | 0.837 |
| Hepatitis | 0.884, 60, 0.632 | 0.89 |
| Nursery | 0.696, 77.09, 0.875 | 0.578 |
| SPECTF Heart | 0.802, 60.3, 0.114 | 0.779 |

**Table 9.** The result for homogenous granulation (*HG*) and for epsilon homogenous granulation ($\varepsilon - HGS$); $5 \times CV5$; $HG\_acc$ = average accuracy for *HG*, $\varepsilon - HG\_acc$ average accuracy for $\varepsilon - HGS$, $HGS\_size$ = *HG* decision system size, $\varepsilon - HGS\_size$ = $\varepsilon - HGS$ decision system size, $TRN\_size$ = *training set size*, $HG_T RN\_red$ =reduction in object number in training set for *HG*, $\varepsilon - HGS\_size$ = reduction in object number in training set for $\varepsilon - HGS$, $HG\_r\_range$ = spectrum of radii for *HG*, $\varepsilon - HG\_r\_range$ = spectrum of radii for $\varepsilon - HGS$, $data_1$ = Australian-credit, $data_2$ = German-credit, $data_3$ = Heartdisease, $data_4$ = Hepatitis.

|  | $Data_1$ | $Data_2$ | $Data_3$ | $Data_4$ |
|---|---|---|---|---|
| $HG\_acc$ | 0.835 | 0.725 | 0.833 | 0.88 |
| $\varepsilon - HG\_acc$ | 0.842 | 0.725 | 0.831 | 0.87 |
| $HGS\_size$ | 286.52 | 513.3 | 120.5 | 46.16 |
| $\varepsilon - HGS\_size$ | 274.52 | 503 | 109.4 | 46.2 |
| $TRN\_size$ | 552 | 800 | 216 | 124 |
| $HG_T RN\_red$ | 48.1% | 35.8% | 44.2% | 62.8% |
| $\varepsilon HG_T RN\_red$ | 50.3% | 37.1% | 49.4% | 62.7% |
| $HG\_r\_range$ | $r_u \geq 0.5$ | $r_u \geq 0.6$ | $r_u \geq 0.461$ | $r_u \geq 0.579$ |
| $\varepsilon - HG\_r\_range$ | $r_u \geq 0.571$ | $r_u \geq 0.65$ | $r_u \geq 0.615$ | $r_u \geq 0.579$ |

## 7. Conclusions

In this work, we have the results of experiments for our new granulation techniques; homogenous and epsilon homogenous granulation. The main advantage of this methods is that there is no need of parameter estimation during approximation. The parameters are tuned in an automatic way by lowering the indiscernibility ratio until the granule contains objects from the same decision class. The reduction of the size of the original decision systems is up to 50 percent. In future works, we plan to check the best classification methods for our new approximation algorithms. Additionally, we wonder if tolerating a fixed percentage of objects from other classes in the granule could improve the quality of classification.

**Author Contributions:** Conceptualization, P.A. and K.R.; Methodology, P.A. and K.R.; Software, P.A. and K.R.; Validation, P.A. and K.R.; Formal Analysis, P.A. and K.R.; Investigation, P.A. and K.R.; Resources, P.A. and K.R.; Writing—Original Draft Preparation, P.A. and K.R.; Writing—Review and Editing, P.A. and K.R.; Visualization, P.A. and K.R.; Project Administration, P.A. and K.R. Funding Acquisition, P.A. and K.R.

**Funding:** This work has been fully supported by the grant from Ministry of Science and Higher Education of the Republic of Poland under the project number 23.610.007-300

**Conflicts of Interest:** The authors declare no conflict of interest. The funders had no role in the design of the study; in the collection, analyses, or interpretation of data; in the writing of the manuscript, or in the decision to publish the results.

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
