# Peer review of "Homogenous Granulation and Its Epsilon Variant†"

_computers, doi:10.3390/computers8020036_

Round 1

Reviewer 1 Report

The paper addresses an interesting issue which regards the homogeneous granulation process using rough inclusion. 

They perform some experiments on public datasets to demonstrate the effectiveness of the proposed method.

According to me, the paper should be revised before it can be considered for publication.

My main concerns are:

- It is not clear the contribution of the paper. It is difficult to understand the real contribution of the paper with respect to the state of the art and to previous works of the same authors. Please, explain your contributions and the novelty of this work.

- The structure and writing of the paper should be improved. I like the fact that the paper is not too long, but in some sections, a better and longer explanation of what is proposed could improve the readability. 

- Sections 3-4 are a bit confusing. Please, explain in a clearer way the example and what emerges when applying your technique. The comprehension of the tables is not straightforward. 

Some sections are just a list of the values of the granules, without any comment on the meaning of these.

Author Response

Paper Number: computers-455191

Paper Title: Homogenous granulation and its epsilon variant*

Reply to the First Reviewer

Thank you very much for providing very helpful comments and suggestions.

According to your very helpful comments and suggestions, we have revised our article.

Comment 1:

The paper addresses an interesting issue which regards the homogeneous granulation process using rough inclusion.

They perform some experiments on public datasets to demonstrate the effectiveness of the proposed method.

Reply: Thank you very much for your good comments.

Comment 2:

According to me, the paper should be revised before it can be considered for publication.

My main concerns are:

- It is not clear the contribution of the paper. It is difficult to understand the real contribution of the paper with respect to the state of the art and to previous works of the same authors. Please, explain your contributions and the novelty of this work.

Reply:

According to your very helpful suggestion, we have added further explanations about our contribution vs previous works. We have added text:

Our motivation to perform this research was an idea to determine effective indiscernibility ratio of decision system approximation without its estimation.

The ratio of approximation has influence on original data size reduction.

In our previous methods we had to estimate this parameter reviewing the set of radii from $0$ until $1$.

In the methods proposed in this work we do not have to perform this operation.

The ratio, for particular central object, is chosen in automatic way, by extending it until the set of objects is homogenous in the sense of belonging to the decision class.

Instead of performing granulation process several times, depending on the number of attributes of the object, this process is performed only once solving the problem

of optimal radii search. Our results are showing reduction of training data sets size by up to $50$ percent maintaining the internal knowledge at a satisfying

level which was measured by efficiency of the classification process. The method is simple, has a square time complexity, $U^{2}$ main operations time the scalar $|A|$.

$U$ is the set of objects of decision system, $A$ the set of conditional attributes.

Comment 3:

- The structure and writing of the paper should be improved. I like the fact that the paper is not too long, but in some sections, a better and longer explanation of what is proposed could improve the readability.

- Sections 3-4 are a bit confusing. Please, explain in a clearer way the example and what emerges when applying your technique. The comprehension of the tables is not straightforward.

Some sections are just a list of the values of the granules, without any comment on the meaning of these.

Reply:

According to your very helpful suggestion, we have added further explanations.

In particular we have added the following clarifications in sect 3:

In this section we have a formal definition of the homogenous granulation process.

In plain words, considering the set of samples from decision system, we can try to lower the size of the system by searching for groups of objects similar in a fixed ratio.

aving those sets (the granules) we can cover the original system searching for granules, which represent all the knowledge from original decision system.

n this particular method we form the group of objects, which belong to the same decision class and have the lowest possible indiscernibility ratio.

It means that the similarity of samples is as low as possible until they are in the same class.

The granule according to this assumption can be defined as $g^{homogenous}_{r_u}$ - see the equation below.

and we have added text in Sect 3.1:

In the table \ref{TDS}, we have exemplary training decision system, which we based on while computing homogenous granules defined in previous section.

The decision system $(U_{trn},B,d)$ is the set of resolved problems, useful in modelling automatic decision process. $U_{trn}$ is the set of objects from $u_1$

until $u_{24}$, $B$ is the set of conditional attributes (description of samples) and contains values from $b_1$ until $b_{13}$. $d$ is a decision attribute, which contains the expert decision used for creating the model of the classification. In our case two possible classes exist: $d \in D = \{1,2\}$. Lets explain the process of granules formation.

For given object $u_1$, which belongs to class $1$ we are looking for the objects from the same class starting from the identical objects (similar in degree $1$) until

the objects are indiscernible in smallest possible degree (are the least similar to $u_1$) and all of them are in class $1$.

In case we lower the indiscernibility ratio to much, the objects $r$-indiscernible will not point on the decision class in an unambiguous way.

In our example ratio $0.385 = \frac{5}{13}$ for granule $g_{0.385}(u_1)$ means that the set contain objects, which are identical with the central one ($u_1$) at least on 5 positions.

For instance object $u_1$ and $u_6$ have the following common descriptors: $a_1=0$, $a_6=0$, $a_7=2$, $a_7=2$, $a_9=1$ and $a_{13}=3$.

 In the covering part we are looking for the set of granules, which represent each object from $U_{trn}$ at least once.

and finally we have added in sect 4.:

The only difference according to homogenous granulation described in Sect. \ref{HG-example} is adding a $\varepsilon$ parameter,

which allows us to use a floating point values in the process of granulation. The rest of the techniques are similar.

We deeply hope that our revised paper could meet your requirements.

Thank you very much for your kind help.

Reviewer 2 Report

This paper is concerned with a really effective and simple techniques for decision approximation homogenous granulation and epsilon homogenous granulation. The idea is interesting. Some comments are given as follows.

1.      It is suggested that the research content of this manuscript should be emphasized in the abstract rather than the previous work.

2.      In the introduction, please give a detailed introduction to the background of this study and previous research contributions.

3.      It is suggested that the time complexity of the proposed method be detailed in the manuscript.

4.      Please carefully polish the sentences in the manuscript.

5.      It is suggested that the method proposed in the manuscript be evaluated from the perspective of computational efficiency, and then compared with other methods.

6.      It is suggested that formulas should be symbolized and numbered. For example, line 186.

7.      Manuscript typesetting is confused. Please make typesetting according to the standard of academic papers.

8.      The format of references is confused, and it is suggested that the format should be unified.

Author Response

Paper Number: computers-455191

Paper Title: Homogenous granulation and its epsilon variant*

Reply to the Second Reviewer

Thank you very much for providing very helpful comments and suggestions.

According to your very helpful comments and suggestions, we have revised our article.

Comment 1:

This paper is concerned with a really effective and simple techniques for decision approximation homogenous granulation and epsilon homogenous granulation.

The idea is interesting.

Reply: Thank you very much for your good comments.

Comment 2:

1.      It is suggested that the research content of this manuscript should be emphasized in the abstract rather than the previous work.

Reply: According to your very helpful suggestion, we have reduced the information about our previous works from abstract. Abstract is reduced to the form:

In the era of Big data there is still place for techniques, which reduce the data size with maintenance of its internal knowledge.

This problem is the main subject of research of family of granulation techniques proposed by Polkowski.

In our recent works we have developed new, really effective and simple techniques for decision approximation - homogenous granulation and epsilon homogenous granulation.

There is no need to estimate the optimal parameters of approximation for these methods, because those are set in dynamic way according to the data internal indiscernibility level.

 In this work we have presented extension of the work presented at ICIST 2018 conference.

 We present results for homogenous and epsilon homogenous granulation with the comparison of its effectiveness.

Comment 3:

2.      In the introduction, please give a detailed introduction to the background of this study and previous research contributions.

3.      It is suggested that the time complexity of the proposed method be detailed in the manuscript.

Reply:

According to your very helpful suggestion, we have revised cited background works and add the text:

Our motivation to perform this research was an idea to determine effective indiscernibility ratio of decision system approximation without its estimation.

 The ratio of approximation has influence on original data size reduction. In our previous methods we had to estimate this parameter reviewing the set of radii from $0$ until $1$.

 In the methods proposed in this work we do not have to perform this operation.

 The ratio, for particular central object, is chosen in automatic way, by extending it until the set of objects is homogenous in the sense of belonging to the decision class.

 Instead of performing granulation process several times, depending on the number of attributes of the object, this process is performed only once solving the problem of

 optimal radii search. Our results are showing reduction of training data sets size by up to $50$ percent maintaining the internal knowledge at a satisfying

 level which was measured by efficiency of the classification process. The method is simple, has a square time complexity, $U^{2}$ main operations time the scalar $|A|$.

 $U$ is the set of objects of decision system, $A$ the set of conditional attributes.

Comment 4:

4. Please carefully polish the sentences in the manuscript.

Reply: According to your very helpful suggestion we have proofread the text.

Comment 5:

5.      It is suggested that the method proposed in the manuscript be evaluated from the perspective of computational efficiency, and then compared with other methods.

Reply: The methods are really fast, its square complexity, the homogenous technique is obviously the fastest, because we don't have to try the set of parameters,

 considering the best one. Method work in adaptive way and is fastest among the other previously tested.

We do not collect the data of time because the it works in really fast way for the one examined. We have added information about time complexity.

text:

The method is simple, has a square time complexity, $U^{2}$ main operations time the scalar $|A|$.

$U$ is the set of objects of decision system, $A$ the set of conditional attributes.

Comment 6:

6.      It is suggested that formulas should be symbolized and numbered. For example, line 186.

Reply:

According to your very helpful suggestion we have proofread the text. We have numbered important eq. using equation environment.

Comment 7:

7.      Manuscript typesetting is confused. Please make typesetting according to the standard of academic papers.

Reply: According to your very helpful suggestion we have proofread the text.

Comment 8:

8.      The format of references is confused, and it is suggested that the format should be unified.

Reply: According to your very helpful suggestion we have unified the references.

We deeply hope that our revised paper could meet your requirements.

Thank you very much for your kind help.

Reviewer 3 Report

The paper proposes an approach for decision approximation, namely, homogenous granulation and its epsilon variant. The major problem that the research is trying to solve is determining the optimal indiscernibility ratio, i.e., the optimal decision of classifications of objects. 

The idea of the research is not clearly explained. The presentation of the draft can be improved in the following way:

1. Using symbols, variables without definition or explanation is not a good academic writing style. Readers from all areas may read it when published. For example, in Equation (1), u, v, r, IND(), A, etc. are not explained or defined. 

Note: check this issue throughout your draft. 

2. You should describe the meaning of a concept when introducing it. For example, when introducing the concept of "rough inclusion", you should not simply copy and paste the equation from the reference. You should explain the meaning of the concept, who originally proposed it, who also contributed to it should all be referenced and credited, and why you use it. "e-normalized Hamming metric" is another example. 

Note: check this issue throughout your draft. Your paper is not a list of steps/concepts. 

3. You need to provide a reference for each concept or equation that is not initially proposed by you. 

4. Inconsistency of equations, for example, Equation (6) uses d(u) = d(v), while in Line 64, you use d(u) == d(v), what's the difference between = and == in your draft?

5. Section 6 in Page 10 didn't explain clearly that the evaluations by using KNN show the benefits of your approach. Another question, why the automatic selection of the radius during the granulation process does not need an optimization? It sounds an important statement in your draft. Please prove it.  

6. 

Author Response

Paper Number: computers-455191

Paper Title: Homogenous granulation and its epsilon variant*

Reply to the Third Reviewer

Thank you very much for providing very helpful comments and suggestions.

According to your very helpful comments and suggestions, we have revised our article.

Comment 1:

The paper proposes an approach for decision approximation, namely, homogenous granulation and its epsilon variant.

The major problem that the research is trying to solve is determining the optimal indiscernibility ratio, i.e., the optimal decision of classifications of objects.

Reply: Thank you very much for your good comments.

Comment 2:

The idea of the research is not clearly explained. The presentation of the draft can be improved in the following way:

1. Using symbols, variables without definition or explanation is not a good academic writing style.

Readers from all areas may read it when published. For example, in Equation (1), u, v, r, IND(), A, etc. are not explained or defined.

Note: check this issue throughout your draft.

Reply: According to your very helpful suggestion, we have added all the explanations. In particular, in the section Granular rough inclusions.

And in other parts of the text, the modifications are in bolded font.

Comment 3:

2. You should describe the meaning of a concept when introducing it. For example, when introducing the concept of "rough inclusion",

you should not simply copy and paste the equation from the reference.

You should explain the meaning of the concept, who originally proposed it,

who also contributed to it should all be referenced and credited,

and why you use it. "e-normalized Hamming metric" is another example.

Reply:

According to your very helpful suggestion, we have added explanations. In bolded font.

Comment 4:

Note: check this issue throughout your draft. Your paper is not a list of steps/concepts.

3. You need to provide a reference for each concept or equation that is not initially proposed by you.

Reply:

According to your very helpful suggestion, we have reviewed citations.

Comment 5:

4. Inconsistency of equations, for example, Equation (6) uses d(u) = d(v), while in Line 64, you use d(u) == d(v), what's the difference between = and == in your draft?

Reply:

According to your very helpful suggestion, we have removed double ==, and replaced it with =, it was resulted from the programmer's habits, notion == is used in C++,

but its unnecessary here.

Comment 6:

5. Section 6 in Page 10 didn't explain clearly that the evaluations by using KNN show the benefits of your approach.

Reply:

We have chosen k-nn as referenced classifier, because its simple and easy to implement for the other scientist to compare their work with us. The classifier is not essential thing here, the more important

is lowering lowering the size of decision systems with maintenance of internal knowledge. In other works, we have checked Naive Bayes classier, SVM, set of rough set classifier, weighted voting classifier,

each of them works in proper way with our methods.

Comment 7:

Another question, why the automatic selection of the radius during the granulation process does not need an optimization? It sounds an important statement in your draft. Please prove it.

Reply:

According to your very helpful question, we have added explanations in the text.

The technique needn't optimisation, because the indiscernibility of objects in decision classes is unchangeable for particular data. And for the other, depends of its indiscernibility levels inside classes.

In our previous works we tried to find the optimal parameters of approximation using some additional techniques like double granulation process, but there was still need to check the set of approximations level.

Here the granulation part works in deterministic way. Of course granular reflection can be not deterministic because there can be slight difference during covering process.

We deeply hope that our revised paper could meet your requirements.

Thank you very much for your kind help.

Round 2

Reviewer 1 Report

The authors have addressed all the reviewers' remarks.

Although the scientific contribution is not so original, the paper could be of interest for researchers interested in this specific, narrow research area of information granulation approachers.

Therefore, I suggest to accept it for publication

Reviewer 2 Report

Authors have improved the manuscript in the revised version. However, there exist some mistakes or typoes. Please check the whole paper. And I suggest authors should rewrite the section "Abstract" which is the summary of the manuscript. The motivation still is not clear. Please highlight your motivation and backgound in the section "Introduction".

Author Response

Paper Number: computers-455191

Paper Title: Homogenous granulation and its epsilon variant*

Reply to the Second Reviewer - second round:

Thank you very much for providing very helpful suggestions.

According to your very helpful comments, we have revised our article.

Comment 1:

Authors have improved the manuscript in the revised version. However, there exist some mistakes or typoes. Please check the whole paper. 

Reply: According to your very helpful suggestion, we have proofread the article and corrected found typos.

Comment 2:

And I suggest authors should rewrite the section "Abstract" which is the summary of the manuscript. The motivation still is not clear. Please highlight your motivation and backgound in the section "Introduction".

Reply: According to your very helpful suggestion, we have modified abstract and introduction to emhasise our motivation.

We have added in abstract:

"Real problem in this family of methods was the choice of  an effective parameter of approximation for any data sets. It was resolved by homogenous techniques. "

The background work are the granulation techniques proposed by Lech Polkowski and by the second author of the paper. It was cited in the introduction. Techniques are usefull to reduce the size of decision systems with maintanace of internal knowledge. And main problem in this family of techniques was the fact, that granular radius (indiscernibility ratio usefull in granulation process) have to be chosen in experimental way. We have described in this work the way how to granulate systems without need of parameters estimation.

We deeply hope that our revised paper could meet your requirements.

Thank you very much once again for your very kind help.

Reviewer 3 Report

The paper proposes an approach for decision approximation, namely, homogenous granulation and its epsilon variant. The approach can be used in the big data area to reduce the size of the decision system. 

The overall quality of the paper is improved after the authors added detailed explanations of important concepts as well as the definitions of symbols. However, there might be some improvements to improve the quality of the paper, for example, 

1) section 4.1 only has two metrics. It doesn't make sense to make it an independent sub-section to me;

2) Line 141 isn't a complete sentence; Section 4.2 needs a re-write, it sounds like a copy paste of experimental data.

Author Response

Paper Number: computers-455191

Paper Title: Homogenous granulation and its epsilon variant*

Reply to the Third Reviewer - second round

Thank you very much for providing very helpful comments.

According to your helpful suggestions, we have revised our article.

Comment 1:

The paper proposes an approach for decision approximation, namely, homogenous granulation and its epsilon variant. The approach can be used in the big data area to reduce the size of the decision system. 

The overall quality of the paper is improved after the authors added detailed explanations of important concepts as well as the definitions of symbols. 

Reply: Thank you very much for your good comments.

Comment 2:

However, there might be some improvements to improve the quality of the paper, for example, 

1) section 4.1 only has two metrics. It doesn't make sense to make it an independent sub-section to me;

Reply: According to your very helpful suggestion, we have merge the sections.

Comment 3:

2) Line 141 isn't a complete sentence; Section 4.2 needs a re-write, it sounds like a copy paste of experimental data.

Reply: According to your very helpful suggestion, we have improved the sentence from line 141 and decided to merge subsections in Sect. 4 to avoid the mentioned impression. The techniques are similar in hand examples, but there is substantial difference, because the first one cannot be enough precise for numerical data.

We deeply hope that our revised paper could meet your requirements.

Thank you very much once again for your kind help.

Round 3

Reviewer 2 Report

Authors have revised the manuscript according to my comments. I can the paper can be accepted to publish in the journal.

Reviewer 3 Report

The paper proposes an approach for decision approximation, namely, homogenous granulation and its epsilon variant. The approach can be used in the big data area to reduce the size of the decision system. 

The overall quality of the paper is improved after several versions of improvements. It is ready for publication.

This manuscript is a resubmission of an earlier submission. The following is a list of the peer review reports and author responses from that submission.